# A Revised Protocol for Culture of Airway Epithelial Cells as a Diagnostic Tool for Primary Ciliary Dyskinesia

**DOI:** 10.3390/jcm9113753

**Published:** 2020-11-21

**Authors:** Janice L. Coles, James Thompson, Katie L. Horton, Robert A. Hirst, Paul Griffin, Gwyneth M. Williams, Patricia Goggin, Regan Doherty, Peter M. Lackie, Amanda Harris, Woolf T. Walker, Christopher O’Callaghan, Claire Hogg, Jane S. Lucas, Cornelia Blume, Claire L. Jackson

**Affiliations:** 1Primary Ciliary Dyskinesia Centre, NIHR Biomedical Research Centre, University Hospital Southampton NHS Foundation Trust, Southampton SO16 6YD, UK; j.l.coles@soton.ac.uk (J.L.C.); jt3v12@soton.ac.uk (J.T.); amanda-lea.harris@uhs.nhs.uk (A.H.); woolf.walker@uhs.nhs.uk (W.T.W.); 2School of Clinical and Experimental Sciences, University of Southampton Faculty of Medicine, Southampton SO16 6YD, UK; K.Horton@soton.ac.uk (K.L.H.); P.M.Lackie@soton.ac.uk (P.M.L.); 3Centre for PCD Diagnosis and Research, Department of Respiratory Sciences, University of Leicester, Robert Kilpatrick Clinical Sciences Building, Leicester LE2 7LX, UK; rah9@leicester.ac.uk (R.A.H.); gmw6@leicester.ac.uk (G.M.W.); c.ocallaghan@ucl.ac.uk (C.O.); 4Paediatric Respiratory department, Royal Brompton and Harefield NHS Foundation Trust, Sydney Street, London SW3 6NP, UK; P.Griffin@rbht.nhs.uk (P.G.); c.hogg@rbht.nhs.uk (C.H.); 5Biomedical Imaging Unit, University Hospital Southampton NHS Foundation Trust, Southampton SO16 6YD, UK; p.goggin@soton.ac.uk (P.G.); Regan.Doherty@uhs.nhs.uk (R.D.); 6Respiratory, Critical Care and Anaesthesia, UCL Great Ormond Street Institute of Child Health, 30 Guilford Street, London WC1N 1EH, UK

**Keywords:** PCD, ALI culture, bio-resource, primary nasal epithelium, diagnostics

## Abstract

Air–liquid interface (ALI) culture of nasal epithelial cells is a valuable tool in the diagnosis and research of primary ciliary dyskinesia (PCD). Ex vivo samples often display secondary dyskinesia from cell damage during sampling, infection or inflammation confounding PCD diagnostic results. ALI culture enables regeneration of healthy cilia facilitating differentiation of primary from secondary ciliary dyskinesia. We describe a revised ALI culture method adopted from April 2018 across three collaborating PCD diagnostic sites, including current University Hospital Southampton COVID-19 risk mitigation measures, and present results. Two hundred and forty nasal epithelial cell samples were seeded for ALI culture and 199 (82.9%) were ciliated. Fifty-four of 83 (63.9%) ex vivo samples which were originally equivocal or insufficient provided diagnostic information following in vitro culture. Surplus basal epithelial cells from 181 nasal brushing samples were frozen in liquid nitrogen; 39 samples were ALI-cultured after cryostorage and all ciliated. The ciliary beat patterns of ex vivo samples (by high-speed video microscopy) were recapitulated, scanning electron microscopy demonstrated excellent ciliation, and cilia could be immuno-fluorescently labelled (anti-alpha-tubulin and anti-RSPH4a) in representative cases that were ALI-cultured after cryostorage. In summary, our ALI culture protocol provides high ciliation rates across three centres, minimising patient recall for repeat brushing biopsies and improving diagnostic certainty. Cryostorage of surplus diagnostic samples was successful, facilitating PCD research.

## 1. Introduction

Primary ciliary dyskinesia (PCD) is a rare disease usually inherited as an autosomal recessive condition although autosomal dominant and X-linked cases exist [1]. The incidence of PCD is approximately 1:10,000, higher in consanguineous populations [2], and it is associated with impaired function of motile cilia in the airways, embryonic node, and reproductive system [3]. This causes a spectrum of symptoms including unexplained neonatal respiratory distress, persistent wet cough from infancy, repeated respiratory infections, rhino-sinus disease, organ laterality abnormality and subfertility [4]. Early diagnosis is essential to initiate treatment, with the aim of slowing disease progression and improving quality of life [5]. 

There is no “gold standard” diagnostic test for PCD [6]. European Respiratory Society and American Thoracic Society guidelines both recommend a multidisciplinary approach using a combination of tests to make a diagnosis [7,8,9]. Ex vivo nasal or bronchial samples obtained by brushing or curette biopsy are imaged by high-speed video microscopy analysis (HSVA) [10,11] and ciliary motility analysed as a frontline functional test [12]. Transmission electron microscopy (TEM) is used to assess and quantify ultrastructural abnormalities of motile cilia [13,14]. Immunofluorescence labelling (IF) can demonstrate the absence or mis-localisation of ciliary proteins [15,16], particularly helpful in cases where no TEM abnormalities are detected such as with DNAH11 [17], DNAH9 [18] and *HYDIN* gene mutations [19]. Genotyping can detect pathogenic bi-allelic or X-linked hemizygous mutations in 50 PCD-related genes to confirm the diagnosis in approximately 70% of well characterized cases [1,3,20]. However, there are still many individuals without a genetic diagnosis. Some genetic defects result in subtle ciliary beat pattern abnormalities, which are difficult to differentiate from secondary defects (e.g., *GAS8* [21], *DNAH9* [18], *CCDC103* [22,23] mutations) by HSVA and appear normal by TEM. *MCIDAS* [24], *CCNO* [25] and *FOXJ1* [26] mutations cause a lack of cilia rather than dyskinesia, and this could be mistaken for severe secondary epithelial damage.

### 1.1. How Can Cell Culture Be Used in PCD Diagnostics and Research?

Secondary damage of cilia caused by infection, inflammation or sampling trauma is a common feature in ex vivo airway samples [27,28,29]. In addition, sample yield may not be sufficient to support the growing array of tests required to diagnose difficult cases. Therefore, to address poor quality, low yield or expanded diagnostic testing, cell culture can be used.

Airway epithelial cell monolayer mini-culture methods have been used to reduce secondary ciliary dyskinesia in the nasal brushings (from chronic sinusitis patients) to enable better ciliary function measurements by HSVA following 3 days in culture with 83% culture success and improved ciliary beat pattern visualization [30]. However, following recent infections, the airway cilia may be shed and this culture method does not allow for cilia re-growth. 

Likewise, monolayer-suspension methods, initially developed for whole resected tissue [31], have been adapted for nasal brush biopsy [32] and are useful for reducing secondary abnormalities in culture for PCD diagnostics. Briefly, nasal brushing cell suspensions are seeded onto a 1% collagen gel substrate and non-adherent cell aggregates are harvested at 24 h for suspension culture with continuous movement. After 24–48 h, cell spheroids form, which are cultured for at least 21 days before analysis of newly formed cilia on the apical surface of the spheroid. This method confirmed a PCD diagnosis or resolved secondary defects in 46 of 59 cultures (78%) [32]. Pifferi et al., also described a later study in which of 151 subjects, a PCD diagnosis could be confirmed in 36 patients using the suspension model optimally following a 5-day culture process [33]. 

Marthin et al., (2017) reported a more rapid nasal epithelial cell culture method, where spheroids spontaneously formed from terminally differentiated nasal epithelium, retaining their original cilia [34]. Spheroids formed in 82% of 18 samples, with the median number of days to harvest being 4 (1–5) in 7 healthy volunteers and 2 (1–5) in 8 PCD patients’ samples. Whilst retaining their original ciliary beat pattern and frequency, spheroids survived up to 16 days (albeit *n* = 1) and provided ciliated spheroids for HSVA and IF testing of cilia [34]. However, using this method, spheroid numbers are not expanded and are limited by the original sampling yield, which may not support a multitude of tests. Additionally, unhealthy or unciliated samples cannot be re-grown to resolve secondary damage since new cilia are not formed. 

Alternatively, basal epithelial cells can be expanded in submerged culture before cells are differentiated on porous membrane inserts within a culture well at an air–liquid interface (ALI) to stimulate cell polarisation and widespread ciliogenesis, which takes 4–6 weeks in vitro. Re-analysis of cilia function after ALI culture gives a further opportunity to carry out HSVA, TEM and IF without secondary health issues [27,35]. 

Cilia regeneration by spheroid suspension or ALI culture can negate the need for patients, with insufficient or inconclusive test results, to undergo repeat nasal brushing biopsy, reduce the time to a diagnosis of PCD and increase the accuracy of HSVA [7,12]. We have also previously shown how ALI cultures may be used as an airway model to investigate nasal epithelial cell interactions with drugs [36,37,38], bacterial [37,38,39,40,41] and viral infections [40,41,42,43]

### 1.2. A Revised Protocol with High Diagnostic Efficiency Is Being Used within the UK PCD Service

Here, we present our ALI culture protocol using commercially available expansion and differentiation media, PneumaCult Ex Plus and PneumaCult ALI (STEMCELL Technologies, Vancouver, BC, Canada). We report its efficacy at three UK diagnostic centres. We also report on creating a bio-resource for PCD research and the performance of cell cultures after cryostorage at University Hospital Southampton (UHS). Due to the current COVID-19 pandemic we have additionally described the protocol modifications at UHS used to mitigate risk of SARS CoV2 infection during patient interaction and sample handling since July 2020. 

## 2. Methods

We collected culture data from 70 consecutive patient samples attending the PCD Centre at University Hospital Southampton (UHS) between 1st April 2018 and 1st April 2019 (Table 1). Local and national R&D and ethical approvals were complied with Southampton and South West Hampshire Research Ethics Committee A (approval reference: CHI395, 07/Q1702/109). We further report data from 128 samples processed at Leicester and 45 samples processed at the Royal Brompton Hospital using the same culture protocol (with some minor variations depending on equipment and consumables).

Patients were diagnosed as “PCD highly-unlikely” if they had a non-suggestive clinical history and/or PICADAR score (a diagnostic clinical prediction rule for PCD) [4], normal nNO, and “PCD unlikely” HSVA on ex vivo nasal brushing. Patients with a suggestive clinical history and/or PICADAR score with inconclusive/insufficient or “PCD likely” HSVA on ex vivo nasal brushings underwent further testing. This included TEM of cilia ultrastructure and IF labelling of ciliary proteins, and could include repeat HSVA following ALI culture and/or genetic testing. See HSVA outcome definitions below. Hallmark TEM or genetics diagnosed PCD and “PCD highly-likely” cases were diagnosed according to the ERS guidelines [3,7,44]. All available diagnostic data were discussed at a multidisciplinary team (MDT) meeting to decide the patients’ outcome and follow up. 

### 2.1. Nasal Epithelial Cell Culture Protocol (with Additional UHS COVID-19 Modifications)

Nasal brushing biopsies were taken from patients’ inferior turbinates, using a 3 mm bronchoscopy cytology brush (Conmed, Utica, NY, USA, #149R). Individually sterile wrapped brushes were opened in clinic and the wire handles were cut to approximately 15 cm to hold (with wire cutters). To maximise chances of a good yield of healthy ciliated tissue, brushings were performed when patients had been free from respiratory exacerbations for at least 6 weeks. Patients were seated (children on parent’s laps) and asked to remove spectacles, and to clear mucus secretions from their nose with a tissue. Clinicians placed one brush sequentially into each nostril (without anaesthetic), ensuring that the patients’ nostrils were clear. Whilst applying gentle pressure to the inferior turbinate the brush was passed back and forth for approximately 5 s, whilst turning, to ensure coverage. The sample was then placed into a capped 5 mL round bottomed cytology tube (Fisher Scientific, Hampton, NH, USA, #10186400) containing 1.5 mL Medium 199 (Fisher Scientific, Hampton, NH, USA, #22350029) supplemented with 1% penicillin (5000 U/mL)/streptomycin (5000 µg/mL) (Fisher Scientific, Hampton, NH, USA, #15070063). 

Since the COVID-19 pandemic, we have adopted several clinical and laboratory modifications to mitigate the risk of acquiring SARS CoV2 infection at UHS, which were approved by UHS and University of Southampton Health, Safety and Risk directorates. We recommend that all those wanting to undertake nasal epithelial cell culture during the COVID-19 pandemic consult their local institute and government policy to ensure health and safety measures are in place. Patients are seen within 48 h of a negative SARS CoV2 polymerase chain reaction (PCR) test. Patients are contacted directly before attending clinic to ensure that they had none of the main COVID-19 symptoms (as cited by http://www.nhs.uk/conditions/coronavirus-covid-19/symptoms/#symptoms including “a high temperature, a new continuous cough and a loss or change to sense of smell or taste”). The clinicians wear full personal protective equipment (PPE) including clinical scrubs with disposable aprons, gloves, FFP3 ‘filtering face piece’ respirator mask (personally fitted) and plastic face shields. The clinical area is ventilated and the number of personnel and patients are restricted to ensure 2 m distancing before and after brushing sampling. To reduce risk of aerosol generation, patients wear disposable surgical face masks whilst in the hospital. Patients enter UHS via a patient-only entrance and are temperature tested on arrival, and escorted to the clinical area directly before their appointment (to minimize interactions with other staff and patients whilst on site). Patient samples are contained and transported at room temperature (within an hour of sampling) to the onsite laboratory within a plastic sample bag within an anti-crush container. Couriered samples from other hospitals are not refrigerated and can take up to 3 h to arrive. Samples received at our containment level 2 laboratory are handled within a class 2 microbiological safety cabinet (MSC). Specific to COVID-19-modified processing: 100 µL of the sample is sent for a SARS CoV2 PCR test via the Public Health England laboratory at UHS and 900 µL of the sample is kept at 4 °C until the test result is returned (24–48 h). Only those samples proven negative are cultured. We have so far not received a nasal brushing sample that has tested positive for SARS CoV2 by PCR. To remove mucus, the remaining 500 µL of sample is washed in 2 mL HBSS without calcium and magnesium, (Gibco, Thermo Fisher Scientific, Waltham, MA, USA, #10532003) and epithelial cells pelleted by centrifugation at 400× *g* for 7 min. Cell pellets are resuspended in 500 µL Medium 199 (including 1% pen/strep) for HSVA (100 µL), IF diagnostic testing (20 µL/slide, *n* = 10) and the remaining 200 µL is fixed in 4% glutaraldehyde for TEM processing.

To remove mucus, cell suspensions (directly from the brush) intended for culture were washed in 2 mL HBSS without calcium and magnesium, (Gibco, Thermo Fisher Scientific, Waltham, MA, USA, #10532003) and epithelial cells pelleted by centrifugation at 400× *g* for 7 min. Cell pellets were resuspended in 1 mL PneumaCult Ex plus medium (STEMCELL Technologies, Vancouver, BC, Canada, Ex plus kit #05040) supplemented with hydrocortisone (0.1%) (STEMCELL Technologies, Vancouver, BC, Canada, #07925) to directly seed cell clusters in 1–2 collagen (0.3 mg/mL, PureCol 5005.B CellSystems, Troisdorf, Germany)-coated wells of a 12-well culture plate (Corning Life Sciences, Corning, NY, USA, #3548). We did not digest or quantify cell numbers directly from the brush biopsies, and in our experience cell health is more important than yield. Under-seeding is to be avoided particularly if cell health is considered compromised after microscopic assessment. Cells were cultured in 37 °C incubators with 5% CO_2_ and ~100% relative humidity and all culture medium contains additional 1% penicillin (5000 U/mL)/streptomycin (5000 µg/mL) (Fisher Scientific, Hampton, NH, USA, #15070063) and 0.002% nystatin suspension (10,000 U/mL) (Fisher Scientific, Hampton, NH, USA, #15340029). Medium was replaced 3 times weekly and cells were passaged with 0.25% trypsin EDTA (Gibco, Thermo Fisher Scientific, Waltham, MA, USA, #11560626) when at 50–70% confluence for both initial seeding passage 0 and at passage 1 in collagen-coated T25 cm^2^ flasks, to give a final basal epithelial cell yield of 1–2 million. All centrifugations were at 400× *g* for 5 min and cells washed twice in 7 mL HBSS to remove residual trypsin (without use of a trypsin inhibitor). At passage 2,100,000 cells were seeded per collagen-coated 12 mm Transwell^®^ with 0.4 µm pore polyester membrane insert (Corning Life Sciences, Corning, NY, USA, #3460) in a 12-well culture plate. Cells on Transwell^®^ inserts were initially cultured submerged in 250 µL PneumaCult Ex plus medium on the apical side and 650 µL of the same medium on the basolateral side. When a confluent monolayer of basal cells was observed (usually between 1 and 2 days) cells were taken to air–liquid interface (ALI) by apical medium removal and replacement of the basolateral medium with 650 µL PneumaCult ALI medium (STEMCELL Technologies, Vancouver, BC, Canada, ALI kit #05001) supplemented with hydrocortisone (0.5%) and heparin (0.2%) (STEMCELL Technologies, Vancouver, BC, Canada, #07925 and #07980, respectively), replaced 3 times weekly and with apical cell washing (briefly with 100 µL HBSS) aspirated to prevent a build-up of mucus. Cultures were harvested between 3 and 6 weeks to allow for optimal ciliation.

### 2.2. Bio-Resource

Between March 2018 and August 2020 UHS has cryopreserved surplus diagnostic cells from 181 PCD clinic patient samples and 30 healthy donor samples. Surplus cells from passage 1 were frozen 1 million per cryovial in 1 mL CryoStor^®^ cell cryopreservation medium (Sigma, St. Louis, MO, USA, #C2874). Cells were initially frozen at −80 °C (graduated freezing −1 °C/minute in a Mr. Frosty^TM^ container Thermo Fisher Scientific, Waltham, MA, USA, #5100–0001), then transferred to liquid nitrogen for longer-term storage. After thawing, washed cells were seeded for research in a smaller Transwell^®^ insert format in 24-well plates. Briefly, 50,000 cells per collagen-coated 6.5 mm Transwell^®^ with 0.4 µm pore polyester membrane insert (Corning Life Sciences, Corning, NY, USA, #3470) in 100 µL PneumaCult Ex plus medium supplemented (apical side) and 350 µL of the same medium on the basolateral side. Cultures were taken to ALI after 1–2 days replacing only the basolateral medium with 350 µL PneumaCult ALI medium supplemented and maintained as detailed above. 

### 2.3. Post-ALI Culture High-Speed Video Microscopy Analysis

Motile cilia were usually first observed by day 7 post-ALI by low power light microscopy with normal ciliary function and widespread coverage confirmed by HSVA [12] from day 20. Cultures were analysed between 3 and 6 weeks post-ALI when considered optimally ciliated. Cultures with no discernible cilia were analysed 6 weeks post-ALI, to examine for static cilia or ciliary aplasia. For HSVA, cells were scraped gently from the membrane with a pipette tip, washed and centrifuged to reduce mucus, transferred in 100 µL PneumaCult ALI medium into a 0.5 mm-depth Coverwell imaging chamber (Sigma, St. Louis, MO, USA, #635051) and mounted on a glass slide. 

Ciliary beat pattern (CBP) and ciliary beat frequency (CBF) were analysed during HSVA using a ×100 objective lens, with samples equilibrated to 37 °C. Videos were recorded at 500 frames per second and analysed at 30–60 frames per second from a minimum of 6 strips of ciliated epithelium as previously described [27]. Observers with extensive experience [45] in ciliary function analysis then denoted the sample as either “PCD likely”, where a widespread “hallmark” beat pattern was observed that was unlikely to be caused by secondary factors alone; “PCD unlikely” where normal ciliary function was observed in at least six areas and any minor abnormalities present could be attributed to obvious secondary factors; “inconclusive” where abnormal ciliary beating was observed which was likely to be due to secondary factors but PCD could not be excluded; or “insufficient” where the quality or quantity of ciliated epithelium was not sufficient for an accurate decision to be made.

### 2.4. Fast Fourier Transform Analysis of Cilia Coverage

We pseudo-quantified the percentage area of cilia coverage on ALI cultures in-situ on a representative subset of 10 consecutive UHS ciliated samples where cilia were motile (cilia coverage on static cultures requires alternative imaging approaches, e.g., by immunofluorescence labelling or SEM not discussed here). HSVA was carried out at 37 °C with a ×20 objective lens (to acquire larger area mean CBF measurements), imaging every 3rd field of view across the midline of each 12 mm Transwell^®^ insert to collect non-overlapping representative data across the whole membrane. Fields of view with significant moving particulates or mucus debris were avoided. Fast Fourier transform analysis of HSVA.cih Photron video files was performed using an in-house written plugin for https://imagej.net/(by Dr Peter Lackie) to determine the proportion of movement to non-movement (CBF in Hz) detected in the video, using a minimum box size of 4 × 4 pixels (Figure 1). The percentage area of movement detected within 16 fields of view was averaged to give a surrogate for cilia coverage. Of 10 samples the mean cilia coverage was 38.9%, which was within the expected range of 15–50% reported in the respiratory tract in vivo [46].

### 2.5. Trans-Epithelial Electrical Resistance Measurements

One hour before trans-epithelial electrical resistance (TEER), measurements were taken, PneumaCult ALI medium was replaced on the test ALI cultures (ciliated at passage 2) (using 100 µL PneumaCult Ex plus medium on the apical side and 350 µL on the basolateral side in the Transwell^®^ insert format in 24-well plates) and incubated at 37 °C. A control well with an empty Transwell^®^ insert containing only medium (no cells) was also prepared. A World Precision Instrument EVOM2 epithelial Volt/Ohm meter with STX2 electrode (chopstick probe) (Fisher Scientific, Hampton, NH, USA, #15169112) was used. The chopstick probe was sterilized in 70% industrial methylated spirit for 5 min and rinse in medium before and after use and between test and control wells. The mean of three measurements was taken and background control measurements were subtracted before calculating mean Ω.cm^2^ (±SD).

### 2.6. Immunofluorescence Labelling of Ciliated Ali Cultures

The membranes of ciliated ALI cultures were excised at 4 weeks using a surgical scalpel blade (15) and placed into the well of a 24-well plate submerged in 100 µL PneumaCult Ex plus medium. The ciliated epithelial cells were scraped from the membrane surface using a pipette tip into the medium. Then, 20 µL of cell suspension was dropped onto each coated Shandon™ Cytoslides™ (Fisher Scientific, Hampton, NH, USA, #12026689) and allowed to air dry in the class 2 MSC. Once dry, slides were sealed in slide mailer boxes (Fisher Scientific, Hampton, NH, USA, #11719885) and transferred to a −20 °C freezer for storage for up to 4 months. For immunofluorescence labelling, slides were thawed and fixed with 4% PFA for 15 min, washed in PBS with 0.1% triton X-100 (Fisher Scientific, Hampton, NH, USA, #T/3751/08) then blocked with 5% milk powder in PBS-triton X-100 for 1 h. After washing, primary antibodies (anti-RSPH4a 1:200, Atlas Antibodies, Sigma, St. Louis, MO, USA, #HPA031197; anti-alpha tubulin 1:500, Sigma, St. Louis, MO, USA, #T9026) in PBS-triton X-100 were incubated for 2 h at room temperature, followed by washing and secondary antibody (Alexafluor 488, Life Technologies, Carlsbad, CA, USA, #A21121; Alexafluor 594, Life Technologies, Carlsbad, CA, USA, #A11012) incubation at a dilution of 1:2500 in PBS-triton x-100 for 30 min at room temperature. DAPI (300 nM) (Molecular Probes, Thermo Fisher Scientific, Waltham, MA, USA #D1306) was added to the final wash before mounting onto coverslips with Mowiol aqueous mounting media. The slides were kept in the fridge (at least overnight) until imaging using a Leica SP8 laser scanning confocal microscope with Leica Application Suite X software v3.5.5.19976 (Leica Biosystems, Wetzlar, Germany). 

### 2.7. Statistics

Descriptive statistics are presented in Table 1. Normality was checked using the Shapiro–Wilk test. Two sample comparisons were undertaken using the student t-test when normality test passed or Mann–Whitney test when normality tests failed. Matched samples were analysed using the parametric paired student’s t test or non-parametric Wilcoxon test. For multiple comparisons One-Way ANOVA was used for parametric samples or Kruskal–Wallis test for non-parametric samples. Statistical analysis was performed in GraphPad Prism 8 (GraphPad, San Diego, CA, USA). A *p*-value less than 0.05 was considered significant.

## 3. Results

### 3.1. How did the Nasal Epithelial Cell Culture Protocol Improve Diagnostic Accuracy at UHS

Sixty-seven of 70 consecutive UHS patients’ (Table 1) samples were cultured and 64 of those 67 ALI cultures were successfully ciliated (95.5%). Of three samples that were not cultured, two were not suspected of PCD (due to a weak clinical history, normal nNO and “PCD unlikely” HSVA on the ex vivo nasal brushing) and one was diagnosed as “PCD positive” with hallmark TEM (absent outer dynein arms) with static cilia (by HSVA on the ex vivo sample) as determined at MDT meetings and surplus epithelial cells were cryopreserved. 

Of 67 in vitro ALI cultures, only 3 (4.5%) failed (1 insufficient sample, 2 due to infection) and HSVA was performed on the 64 ciliated cultures. The original ex vivo CBP was confirmed in seven normal and six abnormal (PCD likely) ALI-cultured samples (19.4%). Fifty-one original ex vivo samples had an equivocal CBP and ALI culture resolved a normal CBP in 30 (44.8%), an abnormal CBP (PCD likely) in 2 (3%) and remained equivocal in 19 samples (28.4%) (Figure 2). As expected, TEM analysis of ALI cultures demonstrated that normal and PCD hallmark defects were replicated in vitro from representative samples (Figure 3). Example HSVA videos showing the quality of ALI culture after equivocal or abnormal “PCD likely” ex vivo samples are shown in Appendix A

The mean CBF was determined in 57 patients’ in vitro ALI cultures with unambiguous CBP (completely static cilia were recorded with a mean CBF of 0 Hz). Of the 57, three cases (two “PCD highly-likely” and one “equivocal” at MDT outcome) with mostly static cilia and some residually moving dyskinetic cilia in the original ex vivo sample became completely static in culture, which we have reported before [27]. The Shapiro–Wilk test showed that ex vivo (W = 0.99, *p* = 0.81) and in vitro (W = 0.96, *p* = 0.06) sample CBFs were normally distributed only when static samples (three in each group) were excluded. Compared to their matched ex vivo samples (*n* = 57) the mean CBF of patients samples significantly varied after in vitro ALI culture (median: 14.7 Hz ranging from 0 to 18.5 Hz ex vivo; 13.9 Hz ranging from 0 to 17.5 Hz in vitro) (Wilcoxon matched pairs *p* = 0.03) (Figure 3). A mean CBF was not considered meaningful in seven patients’ ex vivo samples caused by a mixed motile ciliary beat pattern, three samples were not cultured and three failed accounting for the remaining without a CBF.

The in vitro ALI cultures’ results were compared by patients’ MDT outcome. Of 55 “PCD highly-unlikely” patients, a normal culture CBP was seen in 37 (67.3%) but remained equivocal in 16 (29.1%) and 2 were not cultured. Of seven “PCD”/“PCD highly-likely” patients, an abnormal “PCD likely” HSVA result was seen in five (71.4%), one was progressing but was deliberately “paused” and frozen due to “hallmark TEM defects” [14] and one patient failed due to infection (Figure 2).

There were eight patients with an equivocal MDT outcome due to inconclusive tests. After all tests, five patients with an unconvincing clinical history were considered to have equivocal diagnostic results at MDT evaluation, with patients advised to seek re-referral if symptoms persisted. Of this group, one patient had a normal CBP, three were equivocal after ALI culture and two failed (insufficiency/infection). Two other patients with normal TEM and “PCD likely” HSVA on in vitro ALI culture are being followed up with genetic testing.

### 3.2. Was the Nasal Epithelial Cell Culture Protocol Reproducible across the UK PCD Service?

This culture method was simultaneously adopted by the Leicester and Royal Brompton PCD Centres. One hundred and five of 128 (82%) cultures from the Leicester PCD Centre ciliated and of these, 21 of 29 (72.4%) insufficient biopsy samples were successfully differentiated at ALI culture for diagnostic use, providing HSVA results where patients would otherwise have been recalled for repeat biopsy. At the Royal Brompton PCD Centre, 30 of 45 (66.7%) ALI cultures ciliated. Therefore, 199 of 239 consecutive ALI cultures ciliated across 3 UK PCD diagnostic sites, giving a combined ciliation success rate of 83.3%.

Since the COVID-19 pandemic, UHS has cultured 44 patients’ samples with protocol modifications to mitigate the risk of acquiring SARS CoV2 infection from nasal brushing samples. This includes storing the nasal cell suspensions in Medium 199 (with 1% pen/strep) at 4 °C for up to 48 h, whilst awaiting the outcome of a SARS CoV2 PCR test. Of the 44 samples cultured during this period, 8 (18%) are still in progress, 28 (64%) ciliated successfully and 8 (18%) failed (3 due to bacteria, 3 due to fungus and 2 as a result of insufficient cell yield).

### 3.3. Can a Bio-Resource Extend Diagnostic Testing and Research?

Studies have been challenged by the limited number of PCD patients’ samples available for research; therefore, since March 2018, UHS has cryopreserved surplus diagnostic cells from PCD clinic patient samples following consent (*n* = 181 to August 2020 including 25 (13.8%) confirmed “PCD positive”/“PCD highly-likely”) and 30 healthy donor samples (20 (66.7%) female; median age 37.4, range 21.4–58.3). We have, to date, recovered 6 confirmed “PCD positive”/“PCD highly-likely” samples, 25 confirmed “PCD highly-unlikely” disease control samples and 8 healthy donors from liquid nitrogen storage, with a ciliation success rate of 100% at ALI culture. Here we report representative data from a subset of the thawed samples (Figure 4). The ciliary beat patterns seen on original ex vivo samples (by HSVA) were recapitulated, scanning electron microscopy demonstrated excellent ciliation, and cilia could be immuno-fluorescently labelled in representative cases after cryostorage and ALI culture differentiation (Figure 4).

The physical barrier properties, measured by TEER at 4 weeks post-ALI, of *n* = 10 thawed “PCD highly-unlikely” 383 Ω.cm^2^ (SD ± 86.5) and *n* = 4 defrosted “PCD highly-likely” 314 Ω.cm^2^ (SD ± 153.9) samples were not significantly different to each other (t test) or to *n* = 4 representative fresh (non-frozen) healthy donor in vitro ALI cultures 299 Ω.cm^2^ (SD ± 97.6) (Mann–Whitney test) (Figure 4). HSVA demonstrated that *n* = 4 representative “PCD highly-unlikely” samples with mean nasal brushing CBF of 17.3 Hz, SD ± 2.9 (ex vivo) resolved a normal CBP with mucociliary clearance with a significantly reduced mean CBF of 12.7 Hz, SD ± 2.5 in matched samples that were ALI-cultured (4 weeks) after liquid nitrogen storage (paired t test, *p* = 0.01); while three of these cultures retained a normal CBP, one culture’s CBF fell just below the UHS normal range (11–20 Hz) [27] (Figure 4). Six “PCD positive”/“PCD highly-likely” samples that had predominantly static cilia (*n* = 4) or uncoordinated and stiff dyskinetic cilia (*n* = 2) on the ex vivo nasal brushing samples retained their abnormal CBP at week 4 of ALI culture in vitro after freezing (CBF not shown). The quality of ALI cultures prepared for HSVA after samples were defrosted from cryostorage compared to their original ex vivo samples are shown in Appendix A

## 4. Discussion

After revisions to our 2014 [27] culture protocol, employing the commercially available STEMCELL Technologies media system, we demonstrate an excellent 95.5% success rate of ALI cultures differentiating cilia with 38.9% cilia coverage for analysis at the UHS PCD centre. Three UK PCD diagnostic centres share protocols [47,48]. Therefore, this revised protocol was simultaneously adopted, enabling us to confirm a combined success of 83.3% of ALI cultures ciliated across our national service. Differences in success rates between our centres may have been due to several factors such as different patient demographics, sampling variability and physical management of samples due to our logistical setups. For example, the UHS benefits from having laboratories on the same site as the patient clinics and we were able to repeat insufficient samples during clinic, whilst the patient was still onsite as needed. Since we have introduced COVID-19 mitigation measures, to limit our risk of acquiring the illness from our patients and samples, we have cultured 44 nasal brushing samples and 28 (64%) are so far ciliated, 8 (18%) are ongoing and 8 (18%) failed since July 2020. This is encouraging that a high success rate can be maintained during the constraints of the current COVID-19 pandemic. Analyses following ALI culture have negated the need to recall patients for repeat nasal brushing biopsy after inconclusive HSVA results due to secondary factors, and/or insufficient sample for analyses [27,35]. The excellent yield of cilia coupled with the speed at which normally functioning cilia generate allow for repeat HSVA analysis and if needed repeat TEM to be reported within diagnostic time constraints, with sample also available for IF labelling of cilia and RNA extraction for further genetic splice variant screening (as well as genomic DNA genetic screening) if required. Results from ALI cultures gave added confidence for patient discharge as “PCD highly-unlikely” when abnormalities resolved in culture, or a diagnosis of “PCD positive” or “PCD highly-likely” to be given when abnormalities persist after culture.

Cell culture is an important tool in the diagnostic pathway but is a lengthy and technically demanding process. A reliable system for obtaining a good yield of healthy ciliated epithelial cells is vital. In our experience, most failures occur early in the culture process due to either a lack of viable epithelial cells from nasal brush biopsy or viral/bacteria/fungal infections. To minimise failure rates, we recommend nasal brush biopsies are taken only from patients free from symptoms of infection for the previous six weeks. Basal epithelial cell survival, proliferation and differentiation in vitro rely on high cell densities [49]. With the patients still in clinic, typically cell yields from biopsies can quickly be checked by low power light microscopy to ensure enough material is present for all tests needed. In the event of a poor yield of healthy cells the patient may be approached for another brushing if tolerated. However, in light of the current COVID-19 pandemic, we are not practicing this to limit our exposure to nasal brushing samples in the laboratory. Although this could potentially be circumvented if microscope equipment can be housed within a customized class 2 MSC. As with all long-term cell culture, infection risks are high, maintaining the cultures in a designated culture facility with limited numbers of experienced users is important to maximise success rates.

Airway cells stored in a bio-resource and cultured at ALI retain their pre-frozen characteristics and provide a model for PCD diagnostic testing and investigating the pathogenesis and treatments of respiratory disease (particularly for rare samples) without needing to recall patients for repeat brushings [36,37,38,39,50]. Exceptionally, 100% of samples recovered from liquid nitrogen successfully ciliated (*n* = 6 “PCD positive”/“PCD highly-likely”, *n* = 25 “PCD highly-unlikely” and *n* = 8 healthy donor samples). In *n* = 4 “PCD highly-unlikely” samples with normal CBP and normal CBF (11–20 Hz at UHS) ex vivo, the CBF of ALI cultures significantly (*p* = 0.01) reduced after freezing despite maintaining normal CBP, either due to low sample numbers or the freezing/defrosting process (Figure 4). The CBF reduction was not pronounced after the ALI culture process in 56 patients’ samples that were not previously stored in liquid nitrogen (Figure 3). We believe this method provides great scope for studies of both healthy and diseased ciliated epithelia, yet we advise for CBF studies that a baseline CBF of ALI cultures derived from cryostored cells be established in-house. However, limitations of the model are the reliance upon commercially obtained culture media, and the undisclosed nature of the culture media components. In summary, we have presented an updated protocol for culture of airway epithelial cells, which has been stable and reliable, with consistently high success rates over the course of one year, based on the experience at three PCD diagnostic centres within a national service. Long-term, our patients will benefit directly from reduced recall for repeat samples, and advances in our understanding of disease phenotype and new treatment efficacy. Bio-resourcing will enable us to participate in national and international networks (https://bestcilia.eu/ and https://beat-pcd.squarespace.com/) that are collaborating to better characterise and treat PCD.

## Figures and Tables

**Figure 1 jcm-09-03753-f001:**
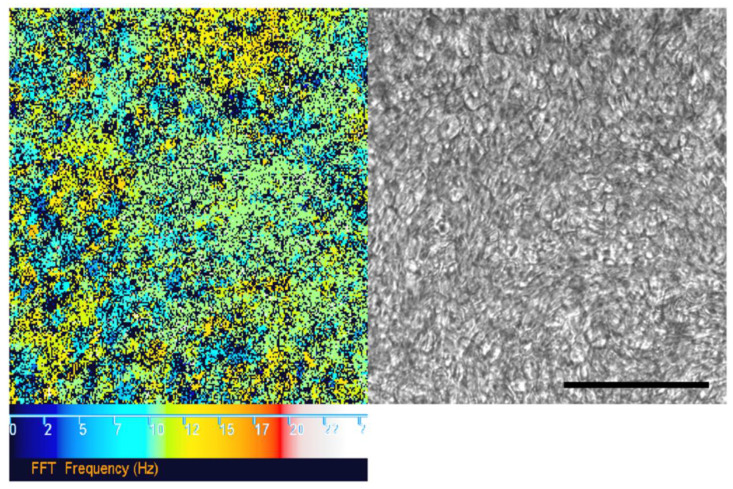
Fast Fourier transform (FFT) analysis of ciliary movement using high-speed video microscopy analysis (HSVA) data. FFT analysis (left) “colour map” corresponding to ciliary movement detected by HSVA (using a 20× objective lens) on the surface of an air–liquid interface (ALI) culture (right). The colour scale (left to right) depicts increasing ciliary beat frequency (CBF) from 0 (black) to 25 Hz (white). Black pixels also represent a CBF measurement outside of the detection threshold (below 2 Hz or above 50 Hz). Normal mean CBF at 37 °C is 11–20 Hz. Scale bar represents 100 µm.

**Figure 2 jcm-09-03753-f002:**
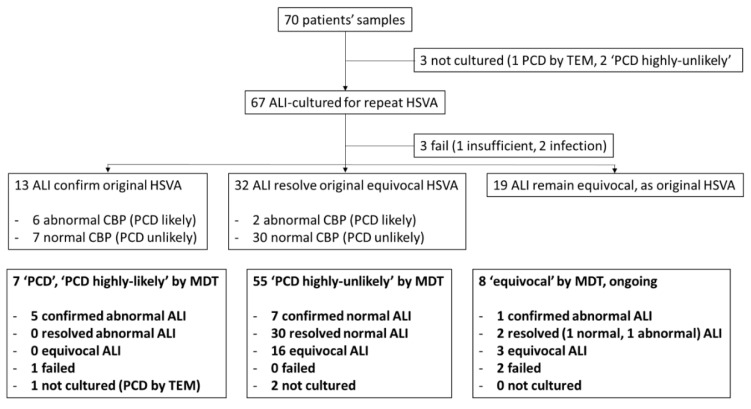
Flow diagram of diagnostic sample processing for ALI culture by repeat HSVA and MDT outcomes. Seventy patients’ ex vivo samples evaluated by HSVA (as part of the whole diagnostic process) and their ALI culture outcomes were followed. HSVA on ALI culture either confirmed the original HSVA finding, resolved an originally equivocal HSVA or remained equivocal despite culture. In bold, the ALI culture HSVA outcomes are shown by MDT outcome (“PCD”/“PCD highly-likely”; “PCD highly-unlikely” or “equivocal” pending follow up, repeat tests or further tests (genetics or additional IF for example).

**Figure 3 jcm-09-03753-f003:**
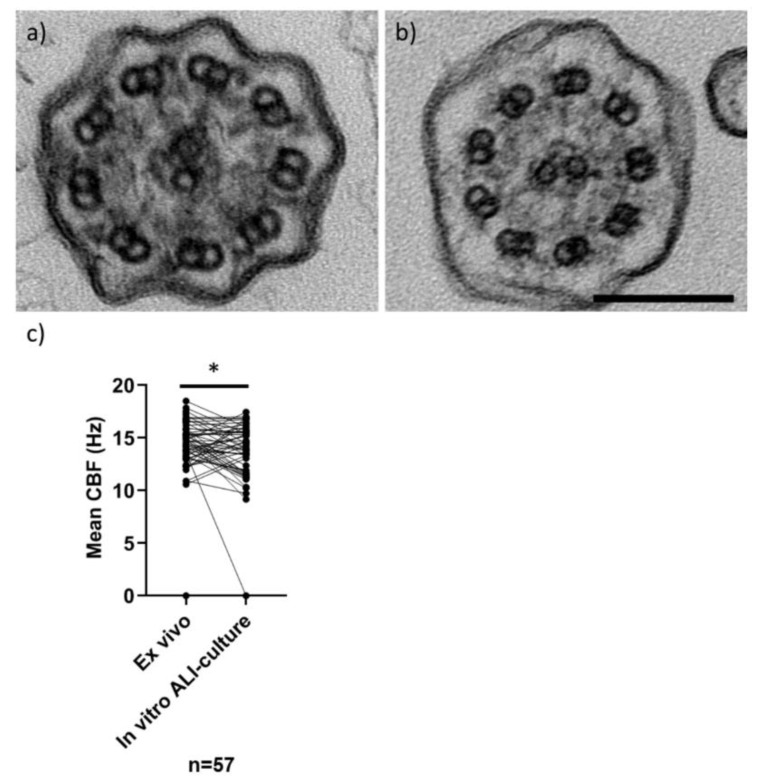
TEM of cilia in transverse section and CBF (by HSVA) in nasal samples and after ALI culture. Representative TEM images (**a**,**b**) of in vitro ALI-cultured cilia in transverse section, showing a “9 + 2” microtubular arrangement. Cilia have normal ciliary ultrastructure in (**a**) a “PCD highly-unlikely” subject (with 3% microtubular defects, 18% inner and 4% outer dynein arm defects quantified from 102 cilia), and (**b**) a “PCD positive” subject (with 11% microtubular defects, 47% inner and 99% outer dynein arm absence quantified from 302 cilia). Scale bar represents 100 nm. Dot plot (**c**) demonstrates the mean CBFs (Hz) of 57 ex vivo nasal brushing samples compared to their matched in vitro ALI cultures (Wilcoxon paired test **p* = 0.03). Data from ex vivo samples without a matched ALI sample were excluded (*n* = 13), which was due to 7 ex vivo samples with a variable ciliary beat pattern (CBP), 3 with failed ALI cultures and 3 that were not cultured. Normal CBF range of ex vivo samples at University Hospital Southampton (UHS) is 11–20 Hz.

**Figure 4 jcm-09-03753-f004:**
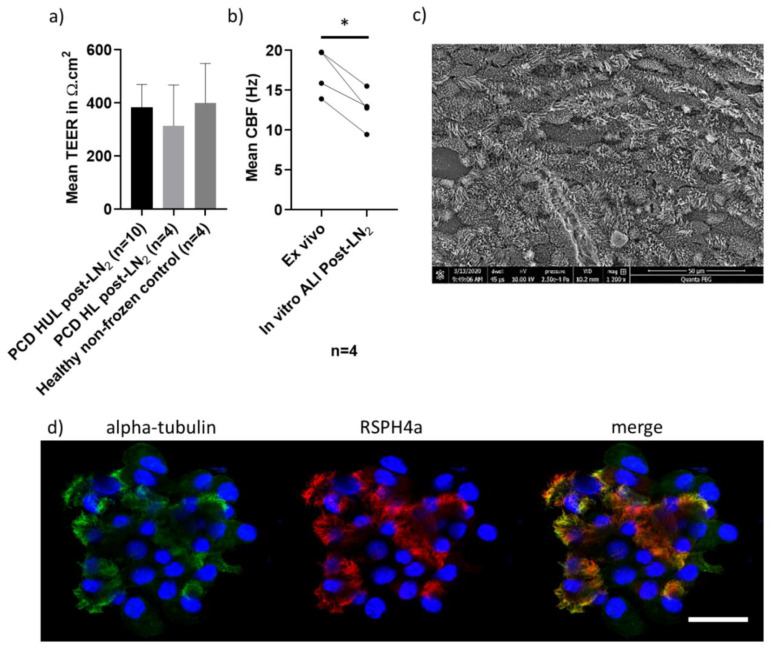
Characteristics of in vitro ALI cultures (ciliated and differentiated at passage 2) derived from frozen liquid nitrogen storage. (**a**) There was no difference in the mean (±SD) trans-epithelial electrical resistance (TEER) (Ω.cm^2^) of *n* = 10 “PCD highly-unlikely” and *n* = 4 “PCD highly-likely” ALI cultures recovered from liquid nitrogen cryostorage (post-LN_2_) (t test), or compared to *n* = 4 healthy donor sample (non-frozen) controls (Mann–Whitney test) measured in triplicate per Transwell^®^ insert at 4 weeks ALI culture, when cells were widely ciliated. (**b**) The mean CBF (Hz) of *n* = 4 matched PCD clinic samples differed before (ex vivo) and after liquid nitrogen storage (in vitro ALI culture) **p* = 0.01 (paired t test). (**c**) A representative SEM image from a “PCD highly-unlikely” ALI culture showing typical ciliation at week 4 post-LN_2_. (**d**) Representative PCD diagnostic immunofluorescence [15] images from an SP8 laser scanning confocal microscope, showing a PCD clinic ALI culture after cryostorage with 4% paraformaldehyde fixation and immunofluorescence labelling with anti-alpha-tubulin (cilia marker-Alexa488 secondary antibody, green), anti-RSPH4a (radial spoke head protein-Alexa549 secondary antibody, red) and DAPI (nuclei DNA stain, blue). Scale 20 µm.

**Table 1 jcm-09-03753-t001:** The characteristics of subjects referred for diagnostic testing to University Hospital Southampton primary ciliary dyskinesia (PCD) centre 1st April 2018 to 1st April 2019, grouped by final multidisciplinary team (MDT) diagnostic outcomes.

	PCD Positive/PCD Highly-Likely (*n* = 7)	PCD Highly-Unlikely (*n* = 55)	Equivocal/Ongoing (*n* = 8)
Median age (min, max)	11.15 (0.1, 32.2)	4.5 (0.1, 70.4)	13.5 (4.0, 63.7)
Female *n* (%)	*n* = 5 (71.4%)	*n* = 26 (47.3%)	*n* = 5 (62.5%)
Chronic wet cough %	100	63.6	87.5
Rhinosinusitis %	85.7	80	75
Situs abnormality %	71.4	12.7	0
Median nNO nl/min	24.4 (Q_1_ 18.8, Q_3_ 47.5, min 18, max 67, *n* = 6)	380 (Q_1_ 240, Q_3_ 596, min 1.2, max 1280, *n* = 49)	280 (Q_1_ 93, Q _3_ 548.5, min 5, max 762, *n* = 8)
TEM	*n* = 3 normal *n* = 2 outer arm defects *n* = 2 outer arm defects with possible inner arm defect	*n* = 49 normal *n* = 6 no data	*n* = 7 normal *n* = 1 no data
Mean ex vivo sample CBF or CBP	*n* = 5 static, 0 Hz *n* = 2 variable dyskinesia	14.8 Hz, SD ± 1.8 (*n* = 52) *n* = 2 variable dyskinesia *n* = 1 insufficient	13.9 Hz, SD ± 1.2 (*n* = 6) *n* = 2 variable dyskinesia
Mean in vitro ALI-culture CBF or CBP	*n* = 5 static, 0 Hz *n* = 2 no data (1 failed due to bacteria, 1 sample frozen)	13.9 Hz, SD ± 2.1 (*n* = 52) *n* = 3 not done	14.4 Hz, SD ± 2.4 (*n* = 4) *n* = 2 became static, 0 Hz *n* = 2 no data (failed due to bacteria)
Causative genes	*RSPH4* homozygous *CCDC151* homozygous *DNAAF3* homozygous *DNAAF5* heterozygous *DNAH11* homozygous *n* = 2 no data	*n* = 13 no mutation found *n* = 42 no data	*n* = 4 no mutation found *n* = 1 *CCDC164* single heterozygous mutation *n* = 3 no data

nasal nitric oxide (nNO); transmission electron microscopy (TEM), ciliary beat frequency (CBF); ciliary beat pattern (CBP).

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
