# Peer review of "A Revised Protocol for Culture of Airway Epithelial Cells as a Diagnostic Tool for Primary Ciliary Dyskinesia"

_jcm, 2020, doi:10.3390/jcm9113753_

Round 1

Reviewer 1 Report

In this article, the authors describe an ALI-culture method for culture airway epithelial cells as a diagnostic tool for primary ciliary dyskinesia. In my opinion, it is a well-conducted work but unfortunately, after reading this manuscript, it is difficult to know where the novelty is or what the authors contribute in relation to this topic. In principle, they described a new protocol for culture airway epithelial cells. Regarding sample collection, they used nasal brushing biopsies to obtain epithelial cells. Although the authors do not provide any reference, this is a standard method commonly used in the literature (Rhinology. 2005 Jun; 43(2):121-4, CHEST. 2010; 138 (6): 1441-1447, Eur Respir J 2013; 41: 960–965, among others) and the authors do not provide any novelty in this regard. The authors used a standard commercial system to culture and differentiate epithelial cells. No references are provided about this system, although many authors have used it. Again they don´t provide any novelty to cryogenization protocols, ciliary motility analysis, etc. The authors state that “Cell culture is an important tool in the diagnostic pathway but is a lengthy and technically demanding process”. This statement has been highlighted by several authors for more than a decade.

This work does not represent a new protocol for epithelial cell culture for PCD diagnosis but one year experience of a PCD diagnosis center and this is how this work should have been oriented. Data presented here is extremely valuable because it helps a better diagnosis of a complex disease such as PCD and my recommendation to the authors is to rewrite this manuscript in this line.

Reviewer 2 Report

The manuscript by Coles et al. describes the use of air-liquid interface (ALI) culture of nasal epithelial cells as a diagnostic tool for primary ciliary dyskinesia. The rationale for this work is based on  confounding PCD diagnostic results by use of ex vivo samples indicate the need for a better and robust airway cell culture model.  A common ALI cell culture was adopted across three collaborating PCD diagnostic sites in the UK. The ALI culture protocol provided high ciliation rates across three centers, minimized patient recall  for repeat brushing  biopsies, and improving diagnostic certainty. A total of 199 out of 239 ALI-cultures were ciliated yielding a combined success rate of 83.3%. However, success rate was lowest at the Royal Brompton PCD Centre (66.7%). Can  authors comment on factors that may have decreased the success rate?

The authors mention that surplus diagnostic cells were cryopreserved from 181 PCD clinic patient samples and 30 healthy donors. 39  out of the 181 frozen PCD samples were  ALI-cultured; all 39 samples developed cilia. Ciliary beat patterns seen on  ex vivo samples (by HSVA) were recapitulated in ALI-cultures, ciliation was confirmed by scanning electron microscopy, alpha-tubulin and RSPH4a expression was detected by immunofluorescence. In addition, transepithelial electrical resistances (TEER) are shown (Fig. 4a) in 18 cultures that were similar among groups but TEER values were relatively low (400 ohmxcm2) and error bars are large. From reading the method section I understand that  TEER must have been measured on passage 2 cells. If this was the case,  please mention in figure legend. The specific method to measure TEER is not described and please add a brief section. Also, comment if  ALI-cultures from TEER measurement could be subsequently used for PCD diagnostics.

The method section for the ALI-cultures lacks detail. The title of the manuscript mentions a protocol for culture of airway epithelial cells, however, little details regarding nasal cell procurement and practical steps involved  are critical for the scientific community to be able to reproduce the culture method. For example, comment if nasal cell procurement was performed by administering a topical anesthetic. What medium/solution was used to collect the brushed cells?  Did you add antibiotics/antifungals to the Pneumacult Expansion and ALI medium? Another variable is the addition of hydrocortisone to the Pneumacult Expansion medium. Did you place cells on ice after collection? Time period between cell collection and cell isolation? Did you enzymatically digest the brushed cells? What was the yield of viable cells? How many cells do you recommend for successful propagation?  You mention to use trypsin for passaging cells, how was the trypsin inactivated? What are your plans and recommendations to address COVID-19 for future sample collections? Please be as detailed as possible to guide the field.

Method for PCD diagnostic immunofluorescence on ALcultures is not provided. Add a detailed protocol to method section. Provide source for antibodies used in the study. At which dilutions were antibodies used?

Minor:

Ln 189: 100 microM should reach 100 micrometer not micromolar

Fig. 4A. Y-axis should be labeled TEER (ohmxcm2). Mention that bars represent mean values in figure legend.

Fig.4B. Visualize normal range. It seems that one point falls below normal range.

Round 2

Reviewer 1 Report

After carefully considering the authors' response to my comments.I would like, first of all, to state the importance of the data presented in the manuscript. In my opinion, these data should be published, but unfortunately, they do not represent a new ALI-culture method for culture airway epithelial cells. In the author response, authors state that “in 2017 there was a catastrophic failure of ALI across all UK centers and many in Europe” and they believe that the culture media was the cause. To demonstrate this, the authors should have conducted experiments culturing biopsies from the same patients using different culture media. Only in this way, the effect of the medium on the success of the culture  can be demonstrated. These experiments have not been performed. My recommendation is to adapt the title and orientation of the manuscript to the data included. I insist, these data is valuable and should be published as the experience of a PCD diagnosis center. Aspects regarding the rate of success of the cultures or aspects concerning culture media should be included in the discussion. I would be pleased to review a new version of this manuscript taken in consideration this proposed changes.

Author Response

Please see the attachment, with thanks.
